# ALP: Action-Aware Embodied Learning for Perception

## Abstract

Current methods in training and benchmarking vision models exhibit an over-reliance on passive, curated datasets. Although models trained on these datasets have shown strong performance in a wide variety of tasks such as classification, detection, and segmentation, they fundamentally are unable to generalize to an ever-evolving world due to constant out-of-distribution shifts of input data. Therefore, instead of training on fixed datasets, can we approach learning in a more human-centric and adaptive manner? In this paper, we introduce **A**ction-Aware Embodied **L**earning for **P**erception (ALP), an embodied learning framework that incorporates action information into representation learning through a combination of optimizing a reinforcement learning policy and an inverse dynamics prediction objective. Our method actively explores in complex 3D environments to both learn generalizable task-agnostic visual representations as well as collect downstream training data. We show that ALP outperforms existing baselines in several downstream perception tasks. In addition, we show that by training on actively collected data more relevant to the environment and task, our method generalizes more robustly to downstream tasks compared to models pre-trained on fixed datasets such as ImageNet.

## 1 Introduction

A vast majority of current vision models are fueled by learning from a passively curated dataset. They achieve good performance on new inputs that are close to the training distribution but fail to generalize across changing conditions. This issue is mitigated partially by performing pre-training on a large dataset like ImageNet or COCO. However, these are still static snapshots of objects which do not allow for richer learning, and pre-trained models still fail to enable generalization to new environments and tasks that are semantically unrelated to those in the large-scale datasets.

A natural approach to address these issues is to leverage *active* agents in an embodied environment. Intuitively, learning representations from exploration in the environment allows us to learn from large scale data that is relevant to the task at hand; active exploration can find maximally informative or diverse instances. A similar argument can be made for active data collection of labeled data for downstream fine-tuning. Furthermore, embodied learning affords the crucial ability to incorporate action information; this is different from learning from static datasets where such signals are not available. As the classic experiment of Held & Hein (1963) shows, the active kitten learns a better visual system despite receiving the same visual stimulation as a passive kitten.

In this paper, we propose **A**ction-Aware Embodied **L**earning for **P**erception (ALP), that leverages action information and performs active exploration both for learning representations and downstream finetuning. In the first stage, we learn a task-agnostic visual representation through a combination of exploration policy and dynamics prediction objectives. Then, in the second stage, for each downstream task, we label a subset of the data collected during the first stage, and finetune our pretrained backbone. See Figure 1 for a schematic of our proposed method.

Importantly, we design our method to incorporate learning signal from interaction - or actions - as follows. We use an inverse dynamics objective during representation learning that explicitly infuses action information into the visual representation by training to predict the action taken to go from one view to another. Similar approaches have been used in prior works such as Agrawal et al. (2016); Jayaraman & Grauman (2015). In addition, we propose a novel idea of using a *shared backbone* for

the representation learning and the reinforcement learning policy for exploration in learning visual perception, while prior works Ye et al. (2021) adopted similar ideas in supervised end-to-end RL tasks. Reinforcement learning indirectly informs learning of visual representations through actions with policy gradients on the exploration reward, by virtue of sharing parameters. This is contrast to methods like CRL Du et al. (2021), which are specifically designed around learning *separate* backbones for the reward and policies models, whereas we demonstrate the benefits of learning in a *coupled* manner, in which all training objectives jointly optimize the same visual backbone.

In summary, our core contributions are as follows:

- We introduce an embodied learning framework that both actively learns visual representation and collects data to be labeled for downstream finetuning. To the best of our knowledge, prior works such as Du et al. (2021); Chaplot et al. (2021) perform only one of the two stages actively.

- We propose to *only* leverage action information from embodied movements for visual representations, implicitly by using a shared backbone for policy and representation learning, in addition to explicit information via inverse dynamics prediction. We experimentally investigate the importance of both kinds of action information through extensive experiments in Gibson across a wide range of tasks (object detection, segmentation, depth estimation).

- We show that our method outperforms baseline embodied learning and self-supervised learning methods, as well as popular pre-trained models from static datasets such as ImageNet. These gains are exemplified on out-of-distribution.

## 2 RELATED WORK

**Visual representation learning from passive data.** Representation learning has shown significant success in computer vision Doersch et al. (2015); Gidaris et al. (2018); Noroozi & Favaro (2016); Pathak et al. (2017b); Wang & Gupta (2015); Zhang et al. (2016); He et al. (2022) by learning generic visual representations through large-scale pretraining, which can then be reused in multiple downstream tasks such as detection and segmentation Cho et al. (2021); Van Gansbeke et al. (2021); Wang et al. (2022); Xie et al. (2021). However, these works only consider representation learning from passively collected datasets and they are fixed snapshots of the world. Instead, we learn visual representation in interactive settings by both actively discovering diverse images and leveraging action information as learning signals for better visual perception.

**Learning perception from active data.** A smaller body of recent work has explored active data collection to improve perception. One line of work focuses on learning pretrained visual representations that are generically good for many downstream tasks. Ye et al. (2021); Ramakrishnan et al. (2021) propose to learn visual representations that can improve sample-efficiency in downstream navigation tasks. Du et al. (2021) learns representations with a min-max game by simultaneously optimizing a contrastive loss in addition to learning a policy that discovers observations that maximizes the same loss. In contrast to our work, these prior works only restrict data collection exclusively to pretraining and do not incorporate underlying action information.

Another line of work develops better heuristics to actively collect samples for specific downstream tasks, such as spatial-temporal inconsistency in semantic predictions Chaplot et al. (2020c), maximizing number of visited objects with high confidence Chaplot et al. (2021), pose information from multiple viewpoints Fang et al. (2020), and encouraging successful interactions with novel objects Nagarajan & Grauman (2020). We instead actively collect training data from intrinsic motivation for both visual representations and downstream tasks in a unified framework. We additionally demonstrate the generality of our learned representations through evaluating on wide ranges of tasks.

**Exploration for Diverse Data Collection.** Outside of representation learning, diversity in training data has been heralded as important and various exploration policies have been proposed to gather diverse data and learn better vision models. Some examples include map building and path planning for environment coverage. Chaplot et al. (2020a) learns to select navigation points from mapping and compute paths to goal location from planning. Another line of work designs intrinsic rewards to measure novelty of state visitations and trains an exploration policy using reinforcement learning

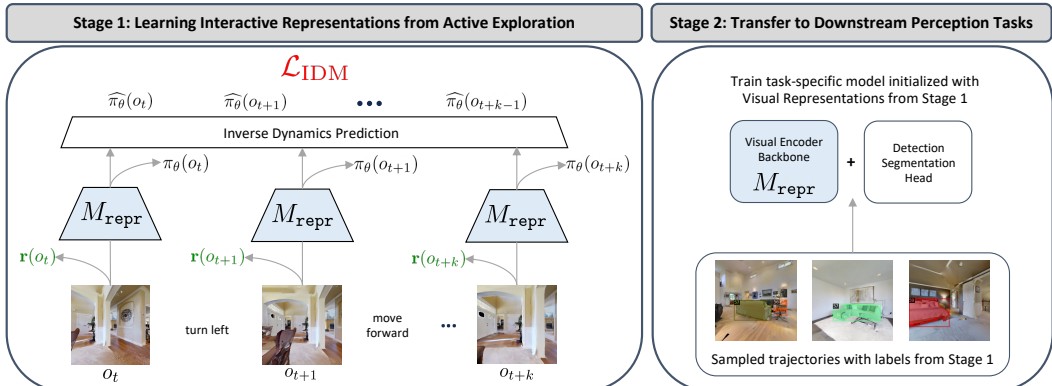

Figure 1: **Overview of ALP.** Our framework consists of two stages. In *Stage 1*, the agent learns visual representations from interactions by actively exploring in environments from intrinsic motivation. Our visual representation learning approach directly considers embodied interactions by incorporating supervisions from action information. In *Stage 2*, we utilize both learned visual representations and label a small random subset of samples from explored trajectories to train downstream perception models.

to maximize rewards Meyer & Wilson (1991); Stadie et al. (2015); Fu et al. (2017); Pathak et al. (2017a); Burda et al. (2019). Our work fall into this category, where performing reinforcement learning to learn a policy allows us to incorporate gradient signals which provide action information into the representation learning (as discussed in Section 3.1).

**Incorporating action information into representation learning.** Previous works that incorporate actions into learned visual representations focus on robot interactions. Pinto et al. (2016); Agrawal et al. (2016); Jang et al. (2018); Pathak et al. (2018); Eitel et al. (2019) leverages dynamics from physical interactions to learn visual representations emerging from physical interactions. While they similarly explicitly use action information, their physical activities are limited to manipulation or poking, we instead allow agents to actively navigate in complex visual environments to collect diverse observations, and use the resulting action information. Furthermore, different from all prior works, we use action information both explicitly through including as part of visual representation learning objectives and implicitly by transferring shared visual representation from policy to downstream perception tasks.

## 3 METHOD

We now present ALP, a general framework to actively collect data for visual representation learning and downstream perception tasks. Figure 1 shows an overview of our method, which consists of two stages. In *Stage 1*, we allow the agent to actively explore in visual environments from intrinsic motivation to discover diverse observations. We propose a coupled approach to incorporate action information from movements and learn a set of shared visual representations that jointly optimizes a policy gradient objective and inverse dynamics predictions loss (Section 3.1). In *Stage 2*, we label a small subset of collected samples from active exploration for downstream perception tasks, and initialize from the learned representation in Stage 1 to improve performance of perception model (Section 3.2). We provide pseudo-code of our framework in Algorithm 1 of Appendix B.1.

### 3.1 ACTION-AWARE EMBODIED LEARNING FOR TASK-AGNOSTIC VISUAL REPRESENTATIONS

As the agent actively navigates in visual environments to gather informative and diverse observations, our goal is to learn visual representations from these sequences of embodied interactions. Such learning signals are not available when training on static datasets. We train a *single network* with the following objectives: (i) policy optimization to encourage visitation of novel states via reinforcement learning (ii) explicitly predicting multiple steps of actions given sequence of observations. We discuss each objective below.

**Learning Exploration Policy.** We use existing novelty-based reward as intrinsic motivation to train exploration policy for our experiments, such as RND Burda et al. (2018) and CRL Du et al. (2021). We provide detailed definitions and equations to compute different rewards we consider in Appendix A. Given computed rewards $\mathbf{r}(o_t)$, we train our policy using Proximal Policy Optimization (PPO) Wijmans et al. (2019); Schulman et al. (2017) algorithm, which optimizes policy $\pi_\theta$ to maximize accumulated sums of rewards over episodes within horizon $T$, given transitions $o_{t+1} \sim \mathcal{T}(o_t, a_t)$ and a discount factor $\gamma$:

$$\max_\theta \mathop{\mathbf{E}}_{a_t \sim \pi_\theta(o_t)} \left[ \sum_{t=0}^{T} \gamma^t \mathbf{r}(o_t) \right] \tag{1}$$

Our representation learning model $M_{\texttt{repr}}$ is updated with Policy Gradient (PG) objective as follows:

$$\mathcal{L}_{\text{PPO}} = \hat{\mathbf{E}}_t \left[ \min \left( r_t(\theta) \hat{A}_t, \text{clip} \left( r_t(\theta), 1 - \epsilon, 1 + \epsilon \right) \hat{A}_t \right) \right] \tag{2}$$

where the estimated advantages $\hat{A}_t$ and expectations $\hat{E}_t$ are computed using value function estimates $\hat{V}(o_t)$, and the clip ratio $r_t(\theta) = \frac{\pi_\theta(a_t|o_t)}{\pi_{\theta_{\text{old}}}(a_t|o_t)}$ is the ratio between the probability of action $a_t$ under current policy and old policy respectively. We use policy gradient to both improve exploration policy and learn visual representations of environments.

A key departure from prior work is to apply the above policy gradient updates on the same network $M_{\texttt{repr}}$ that performs the representation learning. In other words, reinforcement learning is not just to collect diverse data, but to also provide signal on visual features that will be used downstream, since our agent is trained to navigate in photorealistic environments to discover diverse images. Note that since policy gradients involve action information, this loss can also be interpreted as providing indirect signal on the actions. We experimentally find that such action information improves visual representations.

**Explicit Action Prediction.** We make the important observation that active interactions via an embodied agent provides important information beyond just the images themselves — consecutive frames encode agent's movements. We can train the visual representations to capture long-horizon dynamics between frames. We do so via predicting sequence of intermediate actions from sequence of consecutive frames, since it is easier than alternatives such as action conditioned forward frame prediction, given our action space is rather discrete and low dimensional, and already provides substantial gains in practice.

Our Inverse Dynamics Model (IDM) setup consists of our representation learning model $M_{\texttt{repr}}$, a projection head $h_{\texttt{proj}}$, and a prediction network $g_{\texttt{IDM}}$. Given a trajectory consisting of $k$ steps of transition pairs $\{(o_i, a_i, o_{i+1})\}_{i=t}^{t+k-1}$, observations are encoded into visual features $z_i = h_{\texttt{proj}}(M_{\texttt{repr}}(o_i))$. Then these feature encodings of $k+1$ consecutive frames are concatenated as input to the prediction network $g_{\texttt{IDM}}$ to predict $k$ steps of actions taken by the agent. Our predicted distribution of action is approximated by

$$P_{\text{IDM}}(a|o_t, ..., o_{t+k}) = \text{softmax} \left( h_{\texttt{IDM}}(a|z_t, ..., z_{t+k}) \right) \tag{3}$$

Since our action space $\mathcal{A}$ consists of discrete actions, our predicted estimate of action $\hat{a}_i = \text{argmax}_{a \sim \mathcal{A}} P_{\text{IDM}}(a)$; our parameters $M_{\texttt{repr}}$, $h_{\texttt{proj}}$, and $h_{\texttt{IDM}}$ are optimized with cross-entropy loss averaged over $k$ steps:

$$\mathcal{L}_{\text{IDM}} = \mathbf{E}_\tau \left[ -\frac{1}{k} \sum_{i=0}^{k-1} a_i \log P_{\text{IDM}}(a_i|o_t, ..., o_{t+k}) \right] \tag{4}$$

where $\tau = \{(o_i, a_i, o_{i+1})\}_{i=1}^{T}$ denotes a rollout trajectory of length $T$. With supervisions from IDM, our representation learning model $M_{\text{repr}}$ learns visual representations from environments as it is trained to extract features for predicting action movements between consecutive observation frames. In our experiments, our IDM is trained to predict $8$ steps of intermediate action unless specified, since empirically we found this performs better.

Our representation learning model $M_{\text{repr}}$ is optimized with $\mathcal{L}_{\text{PPO}}$ and $\mathcal{L}_{\text{IDM}}$ on every batch of rollouts collected from exploration, both of which provide learning signals from embodied interactions for visual representations.

## 3.2 Transfer to downstream perception tasks

In our framework, the same active exploration process is also used to collect training samples for downstream tasks. The exploration policy for representation learning is aimed at capturing diverse and novel observations from environments. For best downstream performance, we can reuse the same policy to collect informative training data for downstream tasks, and randomly sample a small subset of images periodically and obtain the corresponding semantic labels from the simulator. Given the pretrained representation $M_{\mathrm{repr}}$ from Section 3.1, we initialize the backbone visual encoder of the downstream perception model for each task with pretrained representation $M_{\mathrm{repr}}$ and perform end-to-end supervised learning on labeled samples for each task until convergence.

## 4 Experiments

In this section, we experimentally evaluate ALP and compare various baseline methods for active representation learning, and active downstream finetuning. We particularly focus on understanding gains of incorporating action information — both via using a shared backbone for representation learning and active exploration policy (indirect action signal), and a direct self-supervised loss to predict inverse dynamics. In addition, we perform various ablations to study the importance of different components of our method.

## 4.1 Experimental Setups

**Environments.** We perform experiments on the Habitat simulator Savva et al. (2019) with the Gibson dataset Xia et al. (2018). The Gibson dataset consists of photorealistic scenes that are 3D reconstructions of real-world environments. Identical to previous works Chaplot et al. (2021; 2020b), we use a set of 30 scenes from the Gibson tiny set for our experiments where semantic annotations are available Armeni et al. (2019), with a split of 25 training scenes and 5 test scenes. The list of training and test scenes is provided in Appendix B.2.

**Agent Specification.** Following previous works Chaplot et al. (2021); Du et al. (2021), we allow our agent to receive observations and take actions in a simulator. For each time step $t$, our observation space consists of only $256 \times 256$ RGB observations. Our action space consists of 3 discrete actions: move forward ($0.25m$), turn left ($30°$), and turn right ($30°$).

**Training details.** To pretrain visual representations jointly with learning exploration policy, we allow agents to interact in the environments for around 8M frames; our representation learning model $M_{\mathrm{repr}}$ is optimized using all collected minibatches from explored trajectories. We randomly sample a subset of 10k images with labels per scene from all explored frames as training data for downstream tasks, totaling 250k image-label pairs. We adopt commonly used architectures for each task: for detection and segmentation, we use a Mask-RCNN He et al. (2017) using Feature Pyramid Networks Lin et al. (2017) with a ResNet-50 He et al. (2016) as $M_{\mathrm{repr}}$; for depth estimation, we use a Vit-B/16 encoder-decoder architecture Dosovitskiy et al. (2020) with its encoder as $M_{\mathrm{repr}}$. We provide full hyperparameter, architecture, and implementation details in Appendix B.3.

**Evaluation setups.** Following evaluation setups in previous works Chaplot et al. (2021; 2020b;c), We use 6 common indoor object categories for our experiments: chair, couch, bed, toilet, TV, and potted plant. We consider two evaluation settings:

- **Train Split:** We randomly sample a set of 5000 evaluation images from in-distribution training scenes (200 images per scene).
- **Test Split:** We randomly sample a set of 5000 evaluation images from out-of-distribution test scenes (1000 images per scene). We directly evaluate models on *unseen* environments without further adaptations.

We evaluate downstream perception models trained with ALP and other baselines on each evaluation dataset; we report bounding box, mask AP50 scores, and RMSE for Object Detection (ObjDet), Instance Segmentation (InstSeg), and Depth Estimation (DepEst) tasks respectively. AP50 is the average precision with at least 50% IOU, where IOU is defined to be the intersection over union of the predicted and ground-truth bounding box or the segmentation mask. RMSE is the square root of the mean squared error between the predicted and ground-truth depth annotations.

## 4.2 EXPERIMENTAL RESULTS

### 4.2.1 BENEFITS FROM INCLUDING ACTION FOR PERCEPTION

**ALP improves learning for several exploration methods.** In order to show that our proposed coupled representation learning approach to including training signal from action supervision can generally improve from different active exploration methods, we combine our framework with two learning-based exploration strategies: RND and CRL. For both baseline methods, we follow prior work and train *separate* models to learn the exploration policy and visual representations, and use the same loss (RND prediction error, or CRL contrastive loss) to both incentivize exploration (maximize loss) and learn representations (minimize loss). When combining with ALP, we modify the training process to *share* backbones between reinforcement and representation learning, and add in inverse dynamics predictions losses to provide direct learning signals from actions.

Table 1 reports final performance in perception tasks from each exploration method with and without combining with ALP. We observe significant increase in downstream accuracy when augmenting both RND and CRL with our method (RND-ALP and CRL-ALP, respectively), especially for generalization to test scenes. Interestingly, we see a slightly higher boost when applying ALP to RND Burda et al. (2018) compared to CRL Du et al. (2021), possibly because CRL already learns active visual representations, while RND is specifically designed to measure novelty and our framework learns better visual representation on top of it.

| Method | Train Split | | | Test Split | | |
| --- | --- | --- | --- | --- | --- | --- |
| | ObjDet | InstSeg | DepEst | ObjDet | InstSeg | DepEst |
| CRL | 79.61 | 76.42 | 0.265 | 40.81 | 34.24 | 0.352 |
| CRL-ALP (ours) | **83.14** | **79.24** | **0.261** | **42.42** | **36.89** | **0.350** |
| RND | 83.65 | 81.01 | 0.251 | 37.13 | 34.10 | 0.366 |
| RND-ALP (ours) | **87.95** | **84.56** | **0.237** | **50.34** | **46.74** | **0.305** |

Table 1: **Results of combining ALP with different exploration methods.** We combine our method with two exploration strategies, RND and CRL, and compare performance of downstream perception models with baselines. ALP in most cases improves final performance of perception tasks when integrated with different exploration strategies.

**Action-aware training results in better and more robust visual representations.** We further examine whether applying purely vision-based self-supervised learning objectives on the same pretraining data as RND-ALP is sufficient for accurate downstream performance. In order to test this, we record all explored frames (around 8M) from the agent collected from our method and learn representations using several contrastive learning methods on top of this dataset. Specifically, we consider SimCLR Chen et al. (2020a) initialized from scratch - *SimCLR (FrScr)*, SimCLR initialized from ImageNet pretrained weights - *SimCLR (FrImgNet)*, and Contrastve Predictive Coding Oord et al. (2018) - *CPC*. We report results in Table 2.

Through our proposed representation learning approach that couple training objectives from action supervision, RND-ALP improves much from existing self-supervised contrastive learning methods using the same pretraining and downstream dataset. It even achieves similar performance compared to initializing from ImageNet SimCLR weights, without using any form of contrastive learning. This demonstrates that leveraging action supervisions during embodied movements is comparable to training on much more diverse and curated large-scale data in improving visual representation. We report similar experimental results on CRL-ALP in Table 9 of Appendix C.1.

### 4.2.2 DISENTANGLING THE EFFECT OF VISUAL REPRESENTATION AND DATA COLLECTION QUALITY

Downstream performance of ALP depends on a combination of (1) the quality of the visual representation backbone learned during pretraining, and (2) the quality of the dataset collected and labeled for downstream finetuning. In this section, we control different aspects of our method to gain some

|  | Train Split | | Test Split | |
| --- | --- | --- | --- | --- |
| Method | ObjDet | InstSeg | ObjDet | InstSeg |
| SimCLR (FrScr) | 85.63 | 82.14 | 43.00 | 35.57 |
| SimCLR (FrImgNet) | **90.08** | **86.76** | 47.66 | 44.70 |
| CPC | 86.30 | 83.05 | 41.52 | 37.49 |
| RND-ALP (ours) | 87.95 | 84.56 | **50.34** | **46.74** |

Table 2: **Results of comparing to self-supervised contrastive learning methods.** RND-ALP generally achieves similar or better performance compared to self-supervised learning baselines under same pre-training and fine-tuning data distribution, without including *any* form of contrastive loss.

insight into disentangling the effects of these two factors. For all further experiments, we compare baselines relative to RND-ALP. We report similar results on CRL-ALP in Appendix C.1.

**Comparing visual representation quality.** First, we examine the quality of visual representations that ALP learns. To effectively evaluate this, we finetune several pretrained perception models on *the same downstream dataset* collected from RND-ALP. Our baselines consist of a visual backbones pretrained using *CRL*, *ImageNet SimCLR*, and a from-scratch. We additionally report performance of finetuning *ImageNet Supervised* model as a reference to performance of supervised pretraining.

In Table 3, RND-ALP achieves better performance than our self-supervised learning baselines *only* from images. Since our approach directly incoporates actions to learn visual perception, this demonstrates that such information coming from extensive interactions is more beneficial than pure image-level training, either under embodied settings or on larger-scale curated datasets.

Our framework even performs very close to ImageNet supervised pretraining in our downstream semantic tasks, without access to any class annotations. This is indicates significant benefits from utilizing active interactions for better representation learning, which complements the lack of supervised pretraining. We also perform similar comparisons by fine-tuning various representation learning methods on downstream datasets collected from active Neural SLAM Chaplot et al. (2020a) and passive random policy, and report full results in Table 8 of Appendix C.1.

|  | Train Split | | | Test Split | | |
| --- | --- | --- | --- | --- | --- | --- |
| Method | ObjDet | InstSeg | DepEst | ObjDet | InstSeg | DepEst |
| From Scratch | 86.99 | 83.86 | 0.433 | 38.15 | 34.20 | 0.596 |
| CRL | 87.02 | 82.93 | 0.263 | 48.19 | 42.05 | 0.354 |
| ImageNet SimCLR | 87.22 | 83.50 | 0.255 | **50.66** | 46.55 | 0.352 |
| RND-ALP (ours) | **87.95** | **84.56** | **0.237** | 50.34 | **46.74** | **0.305** |
| ImageNet Supervised | 88.75 | 85.51 | 0.240 | 52.78 | 46.94 | 0.327 |

Table 3: **Results of comparing visual representation baselines.** Performance of finetuning each pretrained visual representation on *the same downstream dataset* collected from RND-ALP. RND-ALP learns better visual representations than baseline embodied learning methods and pretraining from static datasets.

**Comparing downstream data collection quality.** Next, we examine the quality of the downstream data collected by RND-ALP. To accomplish this, we train downstream perception models initialized with *the same pretrained representation* from RND-ALP using different downstream datasets, which are collected from several baseline methods based on active exploration — most notably *RND*, *CRL*, and *Active Neural SLAM (ANS)*.

We report performance of our method and all other baselines in Table 4. Here CRL shows weaker ability to collect better downstream samples possibly because CRL is specifically proposed as an active visual representation leaning method. Compared to ANS as a mapping-based exploration method using path planning, learning-based exploration methods generally achieve better performance under train split; we discuss further in Appendix C.1.

Our results show that RND-ALP improves from baseline active exploration methods by large margins, especially under generalization setting to unseen test split. This suggests that augmented exploration strategies with ALP benefits both visual representations and further dataset quality. We also perform similar comparisons by fine-tuning ImageNet SimCLR and supervised models on each downstream dataset, and report full results in Table 9 of Appendix C.1.

| Method | Train Split | | Test Split | |
|---|---|---|---|---|
| | ObjDet | InstSeg | ObjDet | InstSeg |
| RND | 85.68 | 80.86 | 48.33 | 44.30 |
| CRL | 82.49 | 78.33 | 44.77 | 38.26 |
| ANS | 75.59 | 71.37 | 47.95 | 41.09 |
| RND-ALP (ours) | **87.95** | **84.56** | **50.34** | **46.74** |

Table 4: **Results of comparing downstream data collection baselines.** Performance of finetuning *the same pretrained representations* from RND-ALP on each data collection method. RND-ALP collects better downstream training data compared to several baseline active exploration methods.

### 4.2.3 ABLATIONS

**Contributions of each component to representation learning.** To quantify and understand the importance of each training objective to visual perception, we perform experiments with several ablations. We fix the same exploration process and train another separate visual encoder using either only policy gradient objective or only inverse dynamics prediction loss, then use the same downstream dataset collected from RND-ALP to train the final perception model.

Table 5 shows that both policy gradient and inverse dynamics predictions improve visual representation learning. We observe that training with policy loss generally performs better and thus is more important to learn better visual representations than training with action prediction loss. This further shows that while our exploration agent is trained to navigate to actively discover diverse observations, it also learns visual perception of environments through extensive interactions.

| Pretraining Objectives | | Train Split | | | Test Split | | |
|---|---|---|---|---|---|---|---|
| PG | IDM | ObjDet | InstSeg | DepEst | ObjDet | InstSeg | DepEst |
| ✓ | | **88.60** | 84.36 | 0.239 | 46.56 | 40.34 | 0.334 |
| | ✓ | 86.09 | 83.01 | 0.241 | 42.25 | 37.67 | 0.346 |
| ✓ | ✓ | 87.95 | **84.56** | **0.237** | **50.34** | **46.74** | **0.305** |

Table 5: **Contributions of policy gradient and inverse dynamics.** We train with either policy loss (PG) or action prediction loss (IDM) to evaluate contribution of each component to learned visual representations. Both unlabeled data for representation learning and labeled data for perception tasks are fixed in each of ablation study.

**Number of timesteps in inverse dynamics model.** Since IDM is trained to predict all intermediate actions given a sequence of observations, we wonder whether different training sequence lengths affect the downstream performance. Intuitively, longer training trajectories can capture long-horizon environment dynamics but could be more complex to learn. We report experimental results on RND-ALP in Table 6 and on CRL-ALP in Table 10 of Appendix C.1.

Our results show that IDM trained to predict 8 steps of action sequence generally perform better. It improves from shorter sequence of trajectories possibly since the representation learning model can capture more temporal information, while performance saturates and does not improve further by simply increasing training sequence length.

### 4.3 QUALITATIVE EXAMPLES

In order to qualitatively evaluate how our active agent is able to explore in visual environments, in Figure 2 we visualize trajectories of learned exploration policies in one of environment maps in the Gibson dataset of the Habitat simulator. Compared to baseline exploration agents trained with RND or CRL rewards, RND-ALP and CRL-ALP show wider coverage of environments maps and longer

|                 | Train Split |         |        | Test Split |         |        |
| --------------- | ----------- | ------- | ------ | ---------- | ------- | ------ |
| Number of Steps | ObjDet      | InstSeg | DepEst | ObjDet     | InstSeg | DepEst |
| 4               | **88.25**   | 84.49   | 0.236  | 45.91      | 39.78   | 0.314  |
| 8               | 87.95       | **84.56** | 0.237 | **50.34** | **46.74** | **0.305** |
| 16              | 88.22       | 84.17   | **0.235** | 46.93   | 41.82   | 0.308  |

Table 6: **Number of timesteps in inverse dynamics model (IDM).** We vary lengths of input observation sequence and predicted action sequence when training inverse dynamics model to observe its effect on downstream performance of RND-ALP.

movements trajectory. This demonstrates that ALP improves capability to collect better training samples for downstream tasks. We provide additional examples in Figure 3 of Appendix C.2.

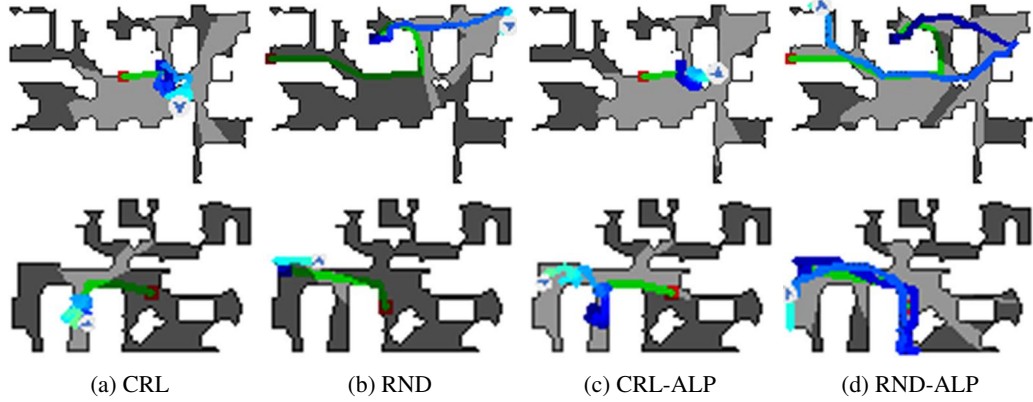

      (a) CRL          (b) RND          (c) CRL-ALP      (d) RND-ALP

Figure 2: Episode trajectories of policies trained with exploration baselines and combined with our method on Habitat environments. We observe that both RND and CRL shows wider coverage of maps and longer movements when combined with ALP.

## 5 DISCUSSION

In this paper, we introduced ALP, a novel paradigm for embodied learning leverages active learning for task-agnostic visual representation and data collection. We demonstrate that using shared visual backbones to incorporate action information through policy gradient and inverse dynamics prediction when pretraining the visual backbone aids in learning better representations that result in more accurate performance across a variety of downstream tasks, such as object detection, segmentation, and depth estimation. Below, we discuss several limitations and directions for future work.

- Although our experiments were primarily done in simulated environments of 3D scans of real locations, a slight domain gap still exists between simulated and real world images. An interesting direction for future work would be to apply ALP to real robots that explore and learn in the real-world, and investigate performance and generalization in these settings.

- We primarily run experiments using a relatively simple action space (forward, left, right). Intuitively, more fine-grained information about actions may better inform an agent about the world and could aid in learning better representations for downstream tasks. As such, a possible direction for future work would be to investigate using our training paradigm in simulators with more diverse action spaces, or on real robots.

- Another avenue for future work would be to explore bootstrapping our learning paradigm using pretrained models or other offline datasets, such as large collections of videos or images scraped from the web. It could also leverage data collected by other policies by pretraining exploration agents with an offline RL method, such as CQL Kumar et al. (2020), then further train online for better active learning. This could help agents more quickly learn to collect diverse data for downstream finetuning.

## REPRODUCIBILITY STATEMENT

We describe experimental setups, implementation details, and hyperparameter values in Appendix B, and include source code as part of the supplementary material.

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

APPENDIX

## A    ACTIVE EXPLORATION WITH INTRINSIC MOTIVATION

Intuitively, our active exploration process in ALP needs to explore the environment to gather informative and diverse observations for both representation learning and downstream tasks. In our experiments, we adopt two existing methods that measure novelty of state visitations as intrinsic rewards and use reinforcement learning (RL) algorithm to train our agent.

**Novelty-based Reward.** To efficiently compute intrinsic reward from high dimensional pixel space, we measure novelty in lower dimensional feature space from visual representation as an empirical estimate. We use a visual encoder $f$ to extract features from RGB observations and use them to compute reward.

Random Network Distillation (RND) Burda et al. (2018) contains a randomly initialized fixed target network $\overline{g}$ and optimizes a trainable prediction network $g$ trained on data collected by the agent to minimize MSE regression error. Intuitively, function approximation error is expected to be higher on novel unseen states and thus encourages the agent to gather diverse observations. Given a visual encoder $f$, its novelty-based reward on observation at timestep $t$ is computed by $\mathbf{r}(o_t) = \|g(f(o_t)) - \overline{g}(f(o_t))\|^2$.

Curious Representation Learning (CRL) Du et al. (2021) designs a reward function specifically beneficial for contrastive loss on visual inputs. Given a family of data augmentations $\mathcal{T}$, its novel-based reward is computed by $\mathbf{r}(o_t) = 1 - \text{sim}\left(g(f(\tilde{o}_t^1)), g(f(\tilde{o}_t^2))\right)$, where $\left(\tilde{o}_t^1, \tilde{o}_t^2\right)$ are two transformed pairs of $o_t$ sampled from transformations $\mathcal{T}$ and $\text{sim}(u, v)$ is the dot product similarity metric. Visual encoder $f$ and projection network $g$ is trained on observations visited by the agent to minimize popular InfoNCE loss Chen et al. (2020a) with data augmentations. Following previous works Du et al. (2021); Chen et al. (2020a), we define $\mathcal{T}$ to consist of horizontal flips, random resized crops, and color saturation using their default hyperparameters.

**Learning Policy.** In practice, we use DD-PPO Wijmans et al. (2019), a distributed version of PPO Schulman et al. (2017) to better scale training in high dimensional complex visual environments. Following previous works Burda et al. (2019), we normalize rewards by the standard deviation of past observed rewards to ensure that reward magnitudes are relatively stable. We train agents with 20 parallel threads, where each environment thread is randomly sampled from training split scenes as specified in Table B.2. The agent is initialized from a randomly sampled location with a random rotation at the beginning of each training episode.

# B    EXPERIMENTAL DETAILS

## B.1    ALGORITHM DETAILS

We present pseudocode describing the process of our ALP framework in Algorithm 1.

## B.2    ENVIRONMENT DETAILS

We provide the list of scenes in train split and test split from Gibson Xia et al. (2018) dataset as follows:

| Train Split | | | | | Test Split |
| --- | --- | --- | --- | --- | --- |
| Allensville | Forkland | Leonardo | Newfields | Shelbyville | Collierville |
| Beechwood | Hanson | Lindenwood | Onagac | Stockmanc | Corozal |
| Benevolence | Hiteman | Marstons | Pinesdale | Tolstoy | Darden |
| Coffeen | Klickitat | Merom | Pomaria | Wainscott | Markleeville |
| Cosmos | Lakeville | Mifflinburg | Ranchester | Woodbine | Wiconisco |

---

**Algorithm 1** ALP

---

**Input:** Environment $E$, Online rollout buffer $\mathcal{B}$, Data buffer $\mathcal{D}$, Intrinsic reward $\mathbf{r}(o)$, Policy $\pi_\theta$ with its visual encoder $M_{\texttt{repr}}$ as representation learning model
**Output:** Representation learning model $M_{\texttt{repr}}$, Downstream training samples $\mathcal{D}$
Initialize $\mathcal{B}, \mathcal{D} \leftarrow \emptyset, \emptyset$
**while** not converged **do**
    *// Collect observations from environments using policy $\pi_\theta$*
    **for** each timestep $t$ **do**
        $o_t = \text{get\_obs}(E), a_t = \pi_\theta(o_t), o_{t+1} = \text{step}(E, o_t, a_t)$
        *// Relabel transitions with intrinsic rewards $\mathbf{r}(o_t)$*
        $\mathcal{B} \leftarrow \mathcal{B} \cup \{\tau_t = (o_t, a_t, o_{t+1}, \mathbf{r}(o_t))\}$
    **end for**
    *// Update representation learning model $M_{\texttt{repr}}$ and exploration policy $\pi_\theta$*
    Update $M_{\texttt{repr}}$ with $\mathcal{L}_{\text{IDM}}(M_{\texttt{repr}}, \mathcal{B})$
    Update $\pi_\theta$ with $\mathcal{L}_{\text{PPO}}(\pi_\theta, \mathcal{B})$
    Update reward network $\mathbf{r}(o)$ with $\mathcal{L}_{\text{reward}}(\mathcal{B})$ (we optimize either RND loss or CRL loss in our experiments)
    *// Collect training samples for downstream tasks*
    **if** record labeled data **then**
        Randomly sample a small subset from current rollout buffer $\{(x_{\text{image}}, y_{\text{label}})\} \sim \mathcal{B}$
        $\mathcal{D} \leftarrow \mathcal{D} \cup \{(x_{\text{image}}, y_{\text{label}})\}$
    **end if**
    Empty online rollout buffer $\mathcal{B} \leftarrow \emptyset$
**end while**

---

### B.3 Implementation Details

To train exploration policy, we use the open-source implementation of DD-PPO baseline Wijmans et al. (2019) from the Habitat simulator `https://github.com/facebookresearch/habitat-lab`. To train downstream perception models, we use the open-source implementation from Detectron2 library `https://github.com/facebookresearch/detectron2` with its default architectures and supervised learning objectives. We provide implementation details of each method and baseline as follows and a full list of hyperparameters can be found in Table 14:

**Implementation details for ALP:**

- **Model architectures.** In exploration policy, following previous works Wijmans et al. (2019), we use a simplified version of agent architecture without the navigation goal encoder. For RGB observations we use a first layer of $2 \times 2$-AvgPool to reduce resolution, essentially performing low-pass filtering and down-sampling, before passing into the visual encoder $M_{\texttt{repr}}$ with output dimension of 2048. This visual feature is concatenated with an embedding of the previous action taken and is then passed into LSTM. The output of LSTM is used as input to a fully connected layer, resulting in a soft-max distribution of the action space as policy and an estimate of the value function.

  In IDM, our projection head $h_{\texttt{proj}}$ is a 2-layer MLP network with an input dimension of 2048, a hidden dimension of 512, and an output dimension of 512; our prediction network $h_{\texttt{IDM}}$ is a MLP with an input dimension of $512 \times$ (num-steps $+ 1$), 2 hidden layers with 512 units, and an output dimension of $3 \times$ num-steps. Then the output layer is chunked by every 3 units, same as dimension of our action space, as predicted logits of IDM.

  We adopt commonly used architectures for each perception task: for object detection and instance segmentation, we use a Mask-RCNN He et al. (2017) using Feature Pyramid Networks Lin et al. (2017) with a ResNet-50 He et al. (2016) as $M_{\texttt{repr}}$; for depth estimation, we use a Vit-B/16 vision transformer encoder-decoder architecture Dosovitskiy et al. (2020) with its encoder as $M_{\texttt{repr}}$.

- **Pre-training details.** To train exploration policy, we use Generalized Advantage Estimation (GAE) Schulman et al. (2015), a discount factor of $\gamma = 0.99$, and a GAE parameter of $\lambda = 0.95$. Each individual episode has a maximum length of 512 steps. Each parallel worker collects 64 frames of rollouts and then performs 4 epochs of PPO with 2 mini-batches per epoch. We use Adam optimizer Kingma & Ba (2014) with a learning rate of $2.5 \times 10^{-4}$. Following previous work Wijmans et al. (2019), we do not normalize advantages as in baseline implementations.

  To train IDM, we use Adam optimizer Kingma & Ba (2014) with a learning rate of $2.5 \times 10^{-4}$. Given 64 frames of rollouts, we perform 4 epochs of gradient updates by using all frames to compute the average prediction loss at each iteration.

- **Fine-tuning details.** To sample a small subset of annotated images from all explored trajectories, we randomly sample 100 annotated images per scene from online batches of rollouts 100 times equally spreaded throughout policy training. We then save them as a fixed static dataset to further train downstream perception models.

  For Mask-RCNN, we use batch size of 32, learning rate of 0.02, resolution of 256 for supervised learning until convergence, roughly around $120k$ training steps and pick the model checkpoint with better performance. For ViT, we use a transformer decoder with 8 attention heads and 6 decoder layers, and train for 10 epochs using the AdamW optimizer with a learning rate of 0.0001 and cosine annealing.

**Implementation details for visual representation learning baselines:**

- **CRL.** Our CRL Du et al. (2021) baseline as visual representation learning method are based on open source implementation available at `https://github.com/yilundu/crl`. For Matterport3D CRL baseline, we use its publicly released pretrained weights in ResNet-50 architecture. For Gibson CRL baseline, we use the same set of hyperparameters as reported in its original setups and pretrain in our environments for 8M frames. We found that given the fixed downstream dataset, pretrained representation from our reproduced experiments in Gibson generally achieve better performance, possibly due to more in-distribution

training data. We thus report all experimental results using our pretrained weights from Gibson environments.

- **ImageNet SimCLR.** For Mask-RCNN results, we tried both SimCLRv1 Chen et al. (2020a) and SimCLRv2 Chen et al. (2020b) checkpoints from `https://github.com/google-research/simclr` with ResNet-50 architecture same as ours and found that SimCLRv2 achieves better performance. We thus report all experiment results from Sim-CLRv2 pretrained weights. For transformers, we pretrain on ImageNet ILSVRC-2012 Russakovsky et al. (2015) with a batch size of 256 for 30 epochs. We use the Adam optimizer Kingma & Ba (2014) with a learning rate of 0.0003 and cosine annealing Loshchilov & Hutter (2016).

- **ImageNet Supervised.** We use ResNet-50 and ViT-B/16 weights pretrained for ImageNet classification task available at `https://dl.fbaipublicfiles.com/detectron2/ImageNetPretrained/MSRA/R-50.pkl` and `https://download.pytorch.org/models/vit_b_16-c867db91.pth`, respectively.

- **Architectures in self-supervised contrastive learning baselines.** For self-supervised contrastive learning baselines compared to ALP, we train a separate network with same architecture as visual backbone in ALP for baseline comparison. For SimCLR, we use a 2-layer MLP projection head with hidden dimensions of 256 and output dimensions of 128 and temperature hyperparameter $\tau = 0.07$ to compute contrastive loss. For CPC, we use a MLP projection head and a forward prediction MLP both with hidden dimension of 256 and output dimension of 128 to compute contrastive loss. We use same learning rate as ALP framework $2.5 \times 10^{-4}$ to optimize contrastive loss.

**Implementation details for downstream data collection baselines:**

- **RND.** Our RND Burda et al. (2018) baseline is based on open sourced implementation available at `https://github.com/rll-research/url_benchmark` and extends to pixel input. We train separate networks for exploration policy and reward model, both with ResNet-50 architecture. For both RND-ALP and RND, we use a 2-layer MLP projection head with hidden dimensions of 512 and output dimensions of 64 to compute reward and to minimize MSE loss. We use the same sampling scheme as in ALP to randomly sample a small subset of annotated images per scene from its explored trajectories as downstream task dataset.

- **CRL.** CRL Du et al. (2021) could also be interpreted as an exploration strategy in addition to an active visual representation learning approach. As proposed in its original setups, we train separate networks for exploration policy and representation learning model, both with ResNet-50 architecture. For both CRL-ALP and CRL, we use a 2-layer MLP projection head with hidden dimensions of 128 and output dimensions of 128 and temperature hyperparameter $\tau = 0.07$ to compute reward and to minimize contrastive loss. We use the same sampling scheme as in ALP to randomly sample a small subset of annotated images per scene from its explored trajectories as downstream task dataset.

- **ANS.** ANS Chaplot et al. (2020a) is a hierarchical modular policy for coverage-based exploration. It builds a spatial top-down map and learns a higher-level global policy to select waypoints in the top-down map space to maximize area coverage. Note that ANS performs depth-based occupancy mapping and assumes sensor pose readings, instead of using only RGB as in our case. Our ANS baseline is based on open sourced implementation available at `https://github.com/devendrachaplot/Neural-SLAM`. For better adaptations, we finetune its pretrained ANS policy in our train split scenes as specified in Table B.2 using its original hyperparameters. We randomly sample a small subset of annotated images from its explored trajectories as downstream task dataset. We generally found that finetuning improves downstream performance of self-supervised visual representations (RND-ALP and ImagNet SimCLR), except that for ImageNet supervised pretrained weights, using pretrained policy achieves better performance (+5.91 in Test Split).

## C  ADDITIONAL EXPERIMENTAL RESULTS

### C.1  QUANTITATIVE RESULTS

**Compare to self-supervised contrastive learning methods.**  We provide additional experimental results comparing CRL-ALP with self-supervised contrastive learning methods on the same unlabeled pretraining and labeled downstream dataset. Note that CRL exploration policy is trained to maximize reward inverse proportional to contrastive loss and thus able to find images that are particularly useful for SimCLR. Table 7 however shows that CRL-ALP improves performance from baseline by only learning from action information.

|  | Train Split | | Test Split | |
| --- | --- | --- | --- | --- |
| Method | ObjDet | InstSeg | ObjDet | InstSeg |
| SimCLR (FrScr) | 77.90 | 75.06 | 40.65 | 36.44 |
| CRL-ALP (ours) | **83.14** | **79.24** | **42.42** | **36.89** |

Table 7: **Results of comparing to self-supervised contrastive learning methods.** CRL-ALP outperforms self-supervised learning baselines under same pretraining and downstream dataset, without including *any* form of contrastive loss.

**Compare visual representation learning qualities.**  We provide additional experimental results comparing our method to visual representation learning baselines. In Table 8 , perception models initialized from all visual representations are trained on the same downstream dataset collected from random policy or from Active Neural SLAM policy. We also include performance of CRL-ALP in addition to RND-ALP.

While ImageNet pretrained model generally performs the best compared to simulator pretraining, ALP still performs better than other simulator-pretrained visual representations. This is significant since we do not use any form of contrastive loss to learn visual representations but only consider different forms of supervisions from active movements.

| Downstream Data | Pretrained Representation | Train Split | | Test Split | |
| --- | --- | --- | --- | --- | --- |
| | | ObjDet | InstSeg | ObjDet | InstSeg |
| ANS | From Scratch | 73.65 | 70.87 | 28.10 | 22.99 |
| | CRL (MP3D) | 77.30 | 73.81 | 40.78 | 36.12 |
| | CRL (Gibson) | 76.60 | 73.10 | 50.72 | 43.79 |
| | ImageNet SimCLR | **77.47** | **74.86** | 47.92 | 41.42 |
| | RND-ALP (ours) | 75.59 | 71.37 | 47.95 | 41.09 |
| | CRL-ALP (ours) | 76.98 | 73.05 | **51.56** | **46.43** |
| | ImageNet Supervised | 80.08 | 76.14 | 43.79 | 36.31 |
| Random Policy | From Scratch | 56.22 | 53.31 | 29.67 | 27.28 |
| | CRL (MP3D) | 58.30 | 53.52 | 33.82 | 30.90 |
| | CRL (Gibson) | 60.52 | 56.39 | 37.45 | 32.12 |
| | ImageNet SimCLR | **64.94** | **61.26** | **43.29** | **37.84** |
| | RND-ALP (ours) | 60.36 | 55.74 | 38.46 | 34.16 |
| | CRL-ALP (ours) | 62.16 | 57.20 | 40.16 | 33.14 |
| | ImageNet Supervised | 65.58 | 62.66 | 43.59 | 39.80 |
| RND-ALP (ours) | RND-ALP (ours) | 87.95 | 84.56 | 50.34 | 46.74 |
| | CRL-ALP (ours) | 88.73 | 84.78 | 49.32 | 43.06 |

Table 8: **Results of comparing visual representation baselines.** Performance of finetuning RND-ALP, CRL-ALP, and all pretrained visual representation baselines on *the same downstream dataset*.

**Compare downstream data collection qualities.** We provide additional experimental results comparing our method with various downstream data collection baselines. In particular, we finetune ImageNet SimCLR or supervised pretrained models and on different downstream data collection baselines. Table 9 shows that RND-ALP collects better downstream data for perception models initialized with different pretrained weights.

Since ANS uses a hierarchical modular architecture and trains a global and a local policy, this is different from training a single policy architecture end-to-end in learning-based exploration methods. We generally found that there are fewer object masks corresponding to samples collected from ANS ($257k$) compared to RND ($284k$) or CRL ($302k$), among $250k$ image-label pairs collected from each exploration policy.

| Pretrained Representation | Downstream Data | Train Split ObjDet | Train Split InstSeg | Test Split ObjDet | Test Split InstSeg |
|---|---|---|---|---|---|
| ImageNet SimCLR | RND | 85.55 | 82.17 | 42.89 | 40.86 |
| | CRL | 82.51 | 79.05 | 42.49 | 38.93 |
| | ANS | 77.47 | 74.86 | 47.92 | 41.42 |
| | RND-ALP (ours) | **87.22** | **83.50** | **50.66** | **46.55** |
| ImageNet supervised | RND | 87.15 | 83.34 | 48.13 | 42.20 |
| | CRL | 84.28 | 80.79 | 46.95 | 41.75 |
| | ANS | 80.08 | 76.14 | 43.79 | 36.31 |
| | RND-ALP (ours) | **88.75** | **85.51** | **52.78** | **46.94** |

Table 9: **Results of comparing downstream data collection baselines.** Performance of finetuning *the same pretrained representations* from ImageNet SimCLR (top) and ImageNet supervised (bottom) on each data collection method.

**Number of timesteps in inverse dynamics prediction.** We provide additional experimental results by varying number of timesteps in IDM when training with CRL-ALP and observe its effects on downstream performance in Table 10.

| Number of Steps | Train Split ObjDet | Train Split InstSeg | Train Split DepEst | Test Split ObjDet | Test Split InstSeg | Test Split DepEst |
|---|---|---|---|---|---|---|
| 4 | 80.63 | 77.30 | **0.259** | **43.97** | **37.09** | 0.352 |
| 8 | **83.14** | **79.24** | 0.261 | 42.42 | 36.89 | **0.350** |

Table 10: **Number of timesteps in inverse dynamics model (IDM).** We vary lengths of input observation sequence and predicted action sequence when training inverse dynamics model to observe its effect on downstream performance of CRL-ALP.

**Variance across multiple runs.** We try to understand variance across multiple runs given *the same* pretrained representation and *the same* downstream data. Specifically, we initialize Mask-RCNN with the pretrained representation and finetune on the labeled dataset both from RND-ALP. Results over 3 runs in Table 11 shows consistent evaluation performance.

C.2 QUALITATIVE EXAMPLES

In Figure 3, we provide additional visualizations of episode trajectories; each line represents policy rollout trajectories in the same house of Habitat environment. We observe that learned exploration policy learned from RND and CRL baseline shows wider coverage of maps and longer movements when combined with ALP, visually demonstrating better exploration policy from ALP framework. We include corresponding videos in the supplementary material.

| Train Split | | Test Split | |
|---|---|---|---|
| ObjDet | InstSeg | ObjDet | InstSeg |
| 87.95 | 84.56 | 50.34 | 46.74 |
| 87.30 | 83.05 | 49.35 | 47.65 |
| 88.19 | 84.06 | 50.79 | 46.36 |
| $87.81 \pm 0.46$ | $83.89 \pm 0.77$ | $50.16 \pm 0.74$ | $46.92 \pm 0.67$ |

Table 11: **Variance across multiple runs.** We report downstream model performance with their mean and standard deviation over 3 runs, given the *same* pretrained representation and the *same* downstream dataset from RND-ALP.

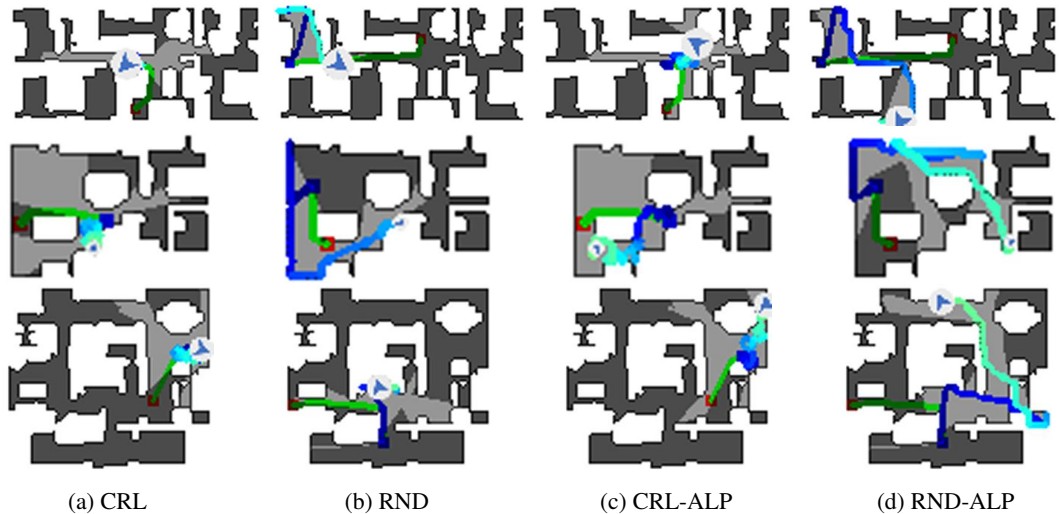

(a) CRL          (b) RND          (c) CRL-ALP          (d) RND-ALP

Figure 3: Episode trajectories of policies trained with exploration baselines and combined with our method on a Habitat environment. Both RND and CRL shows wider coverage of maps and longer movements when combined with ALP.

## D    CONTRASTIVE LEARNING AS PART OF TRAINING OBJECTIVE

We also investigated including contrastive loss as self-supervised visual representation learning objective and provide more details below.

### D.1    METHOD

In principle, we could use any representation learning objective suitable for learning visual representations in a self-supervised manner. In particular, we utilize CPC Oord et al. (2018) when combining with RND exploration method, one of popular contrastive learning approaches Chen et al. (2020a); Stooke et al. (2021), since observations from temporally closer frames should be closer in visual representation space than further frames or observations from different environments. We utilize SimCLR Chen et al. (2020a) with data augmentations when combined with CRL, since this is easier to integrate with its original framework that already introduces a self-supervised loss.

Given our representation learning model $M_{\mathrm{repr}}$ and projection head $h_{\mathrm{con}}$, we encode each observation frame $o$ into its latent representation $z = h_{\mathrm{con}}(M_{\mathrm{repr}}(o))$, and minimize InfoNCE loss as training objective:

$$\mathcal{L}_{\mathrm{contrast}} = -\frac{1}{N} \sum_{i=1}^{N} \log \frac{\exp\left(\mathrm{sim}(z_i, z_{i+})\right)}{\sum_{j=1}^{N} \exp\left(\mathrm{sim}(z_i, z_{j+})\right)}$$

where $\text{sim}(u, v) = v^T u$ is dot product similarity metric between feature vectors. For CPC, positive samples come from temporally close frames within 4 steps and negative samples come from observations in other paralleled training environments. For SimCLR, positive samples come from two augmented pairs of the same image and negative samples come from augmented views from different images.

We modify a few design choices when including contrastive loss in order to improve compute and memory efficiency. To train inverse dynamics model, we optimize cross-entropy loss based on predicted action between two consecutive frames, i.e. $h_{\text{IDM}}(a_t|o_t, o_{t+1})$, instead of training on sequence of observations and actions. The visual encoder $f$ to compute intrinsic rewards is a momentum encoder He et al. (2020) that parameterizes a slow moving average of weights from our representation learning model $M_{\text{repr}}$. We leave more complicated architectures as future investigations.

| Method | Train Split | | | Test Split | | |
| --- | --- | --- | --- | --- | --- | --- |
| | ObjDet | InstSeg | DepEst | ObjDet | InstSeg | DepEst |
| CRL | 79.61 | 76.42 | 0.265 | 40.81 | 34.24 | **0.352** |
| CRL-ALP (ours) | **80.84** | **77.44** | **0.262** | **44.23** | **37.60** | 0.354 |
| RND | 83.65 | 81.01 | 0.251 | 37.13 | 34.10 | 0.366 |
| RND-ALP (ours) | **87.78** | **84.12** | **0.239** | **48.63** | **45.80** | **0.308** |

Table 12: Results of combining ALP (including contrastive loss) with different exploration methods. We combine our method with two exploration strategies, RND and CRL, and compare performance of downstream perception model with baselines.

## D.2 EXPERIMENTAL RESULTS

We perform similar sets of experiments on ALP framework including contrastive loss. We show that including contrastive loss as self-supervised training objective achieves similarly good performance. In Table 12, we report results when combining our framework with two learning-based exploration methods, RND and CRL. We observe much improvements in downstream performance compared to baselines, and are consistent under different exploration strategies. In Table 13, we report full results comparing our framework RND-ALP with visual representation baselines and data collection baselines respectively. RND-ALP outperforms other baselines in downstream perception tasks; it performs similar or even better than ImageNet supervised pretraining without access to any supervisions from semantic labels. Thus this indicates significant benefits of learning signals from active environment interactions in learning better visual representations.

| Pretrained Representation | Downstream Data | Train Split | | | Test Split | | |
| --- | --- | --- | --- | --- | --- | --- | --- |
| | | ObjDet | InstSeg | DepEst | ObjDet | InstSeg | DepEst |
| From Scratch | | 83.50 | 79.65 | 0.327 | 36.05 | 30.57 | 0.524 |
| CRL | | 85.06 | 81.54 | 0.263 | 45.95 | 42.53 | 0.351 |
| ImgNet SimCLR | RND-ALP | 86.38 | 83.40 | 0.257 | 45.26 | 37.07 | 0.349 |
| ImgNet Sup | | 86.99 | **84.28** | 0.244 | 48.53 | 43.40 | 0.340 |
| RND-ALP | | **87.78** | 84.12 | **0.239** | **48.63** | **45.80** | **0.308** |
| | RND | 86.36 | 82.03 | 0.245 | 42.37 | 36.10 | 0.342 |
| RND-ALP | CRL | 83.12 | 79.43 | 0.274 | 40.76 | 34.43 | 0.365 |
| | ANS | 75.35 | 71.73 | - | 48.43 | 42.62 | - |
| | RND-ALP | **87.78** | **84.12** | **0.239** | **48.63** | **45.80** | **0.308** |

Table 13: Results of comparing pretrained visual representations and downstream data collection baselines. We fix either visual representation or labeled dataset from RND-ALP (ours including contrastive loss) and compare downstream performance to all baselines respectively.

| Hyperparameter | Value |
| --- | --- |
| Observation | (256, 256), RGB |
| Downsample layer | AveragePooling (2, 2) |
| Hidden size (LSTM) | 512 |
| Optimizer | Adam |
| Learning rate of $\pi_\theta$ | $2.5 \times 10^{-4}$ |
| Learning rate annealing | Linear |
| Rollout buffer length | 64 |
| PPO epochs | 4 |
| PPO mini-batches | 2 |
| Discount $\gamma$ | 0.99 |
| GAE $\lambda$ | 0.95 |
| Normalize advantage | False |
| Entropy coefficient | 0.01 |
| Value loss term coefficient | 0.5 |
| Maximum norm of gradient | 0.5 |
| Clipping $\epsilon$ | 0.1 with linear annealing |
| Learning rate of reward network | $1 \times 10^{-4}$ |
| Optimizer | Adam |
| Learning rate of IDM | $2.5 \times 10^{-4}$ |
| Optimizer | Adam |
| Number of timesteps | 8 |
| IDM epochs | 4 |
| Projection network | $[512]$ |
| Prediction network | $[512, 512]$ |

Table 14: Hyperparameters for training exploration policy and representation learning model. We follow previous works Wijmans et al. (2019); Du et al. (2021) as close as possible.

