# OpenReview forum: "ALP: Action-Aware Embodied Learning for Perception"
_ICLR.cc/2024/Conference — Submitted to ICLR 2024_

### Official Review · Reviewer_e8e8 · 2023-10-25

**Soundness:** 3 good
**Presentation:** 3 good
**Contribution:** 3 good
**Rating:** 3
**Confidence:** 4

**Summary:**

The paper proposes Action-Aware Embodied Learning for Perception (ALP) which leverages active exploration and action information for representation learning and finetuning for downstream tasks. In the first stage, ALP learns a task-agnostic visual representation by combining exploration policy and inverse dynamics prediction objectives. During second stage a subset of data from first stage is labelled and used for finetuning the pretrained backbone. Authors claim they propose the novel idea of using a shared backbone for representation learning and exploration policy trained with RL. The paper also shows that the proposed method outperforms baseline methods (especially in OOD scenarios) using SSL, embodied learning methods, and pretrained models trained on ImageNet.

**Strengths:**

1. Proposed approach of learning visual representation learning actively using just action aware objectives is interesting and novel
2. The experiment section is thorough and nicely ablates all important questions required to demonstrated effectiveness of ALP.
    1. Table 1 and 2 demonstrate effectiveness of ALP compared to other representation learning methods
    2. Table 3 and 4 ablates importance of representation learning methods by fixing finetuning data and importance of different downstream data collection methods.
3. Ablations in section 4.3.2 clearly demonstrates importance of combining inverse dynamics and RL objectives for training effective visual representations
4. Proposed method outperforms baselines trained using static image datasets like ImageNet
5. Paper is well written and experiments are well organized

**Weaknesses:**

1. The proposed method underperforms compared to SimCLR ImageNet baseline in table 2 on train split and the results for depth estimation are not provided which contradicts the claims made by the authors.
2. As shown in table 3 the results of finetuning different baselines using same finetuning dataset (collected using RND-ALP) the proposed method is only slightly better ~0.2-0.7% compared to ImageNet baseline. It’d be nice if authors run this experiment with multiple seeds and share mean and variance to show the improvements are consistent across seeds.
3. Table 4 compares different data collection methods for finetuning where RND-ALP outperforms existing methods. I’d be interested to see a comparison with a baseline that collects data using random sampling strategy and using a heuristic based exploration agent like a frontier exploration agent which is maximizing the coverage in the environment. The frontier agent baseline is similar to ANS but builds a map using depth + camera info which leads to much better maps for exploration. This can be implement by leveraging codebase from [1]. Based on videos attached in supplementary it looks like RND/CRL/RND-ALP are learning simple navigation behavior which might restrict the diversity of the dataset used for training. It’d be nice to compare the performance with such simple heuristic based data collection baselines
4. In addition to ImageNet pretrained baselines there has been some recent works like MAE, VC-1[2], R3M[3], MVP[4]. Can authors please explain why the comparison with these representation learning baselines are missing? I'd like to see how well ALP does in comparison to these other pretrained representations
5. For each of the tasks ObjDet, InstSeg, and DepthEst can authors also add comparison with the SOTA baselines for each task finetuned on data collected for finteuning? For example, it is important to compare how well a coco pretrained ObjDet model when finetuned on sim data for the 6 categories being used.

[1] Chaplot, D.S., Gandhi, D., Gupta, A. and Salakhutdinov, R., 2020. Object Goal Navigation using Goal-Oriented Semantic Exploration. In Neural Information Processing Systems (NeurIPS-20)
[2] A. Majumdar, K Yadav, S. Arnaud, Y. Jason Ma....... {Where are we in the search for an Artificial Visual Cortex for Embodied Intelligence? NeurIPS 2023
[3] Nair, S., Rajeswaran, A., Kumar, V., Finn, C., and Gupta, A. R3M: A Universal Visual Representation for Robot Manipulation. CoRL, 2022.
[4] Radosavovic, I., Xiao, T., James, S., Abbeel, P., Malik, J., and Darrell, T. Real world robot learning with masked visual pre-training. In 6th Annual Conference on Robot Learning, 2022

**Questions:**

1. The authors claim idea of learning a shared backbone for representation learning and reinforcement learning for exploration policy is novel but I have seen multiple papers that jointly learn visual backbone with RL for end-to-end tasks[1,2]. Can authors please clarify if they meant learning visual representations using RL+IDM only objectives as novelty here? If yes, can the line in the introduction section rephrased?

My major concerns are about multiple seeds for training for experiments in section 3, missing results for depth estimation in table 2 and clarification on why RND-ALP does poorly in table 2, missing comparison with other pretrained representations like MAE, VC-1, etc. In addition if authors can provide comparison with listed data collection strategies in weakness point 3 I’d appreciate that. I am willing to update rating if authors address my concerns

[1] Erik Wijmans, Abhishek Kadian, Ari Morcos, Stefan Lee, Irfan Essa, Devi Parikh, Manolis Savva, and Dhruv Batra. Dd-ppo: Learning near-perfect pointgoal navigators from 2.5 billion frames.
[2]  Joel Ye, Dhruv Batra, Erik Wijmans, and Abhishek Das. Auxiliary tasks speed up learning point goal navigation. In Conference on Robot Learning, pp. 498–516. PMLR, 2021.

---

> ### Author Response · Authors · 2023-11-18
> **Response to Reviewer e8e8 (Part 1)**
>
> We sincerely thank you for helpful feedback and insightful comments. We appreciate that our paper is recognized for several positive aspects: (1) our proposed framework ALP is interesting and novel in that we actively learn visual representations only using action aware training objectives; (2) our experiments and ablations are extensive and thorough, covering aspects related to effectiveness of our framework, and results outperform baselines; (3) paper is well written and organized. We address your comments and questions below:
>
> ---
>
> **Results from multiple random seeds**
>
> We already include experimental results of multiple random seeds for RND-ALP in Table 11 in Appendix. Given the same pre-trained representation and the same task samples dataset, our 3 runs of results achieve similar performance.
>
> We additionally include downstream performance of RND-ALP over 2 runs with different runs for both pre-training and fine-tuning stages below. We generally found that performance in unseen scenarios contains slightly higher variance than that in training environments across multiple runs; overall our method could achieve consistent performance.
>
> | Train Split || Test Split ||
> | :--- | :--- | :--- | :--- |
> | ObjDet | InstSeg | ObjDet | InstSeg |
> | 87.95 | 84.56 | 50.34 | 46.74 |
> | 88.29 | 83.60 | 50.82 | 43.20 |
>
> ---
>
> **Data collection baselines**
>
> We report experimental results of fine-tuning different visual representations in downstream labeled data using random policy in Table 8 of Appendix C. We also present further comparisons with ANS method in the same Table, which tends to achieve stronger performance than frontier-based exploration. We generally found that models trained on passive samples from random policy underperform models trained on active data in both evaluation settings. This demonstrates the importance of active data collection in improving visual representation learning and training better downstream task models.
>
> ---
>
> **Explanation of RND-ALP performance**
>
> For SimCLR (FrImgNet) in Table 2, we initialize with ImageNet SimCLR pre-trained weights, and continue to learn representation from all actively explored frames by RND-ALP using contrastive loss with data augmentation. Our intuition is that ImageNet SimCLR provides a stronger initialization since it sees much more diverse curated data, and together with actively explored diverse samples from the simulator this boosts the representation quality in training environments, while at the same time we generally found that contrastive loss slightly hurts performance in unseen test scenarios. However, we found that ALP outperforms other visual representations trained from-scratch in simulator (Table 2 and 7); this shows action aware training objectives result in performance gains than self-supervised contrastive learning in embodied settings.
>
> ---
>
> **Compare to COCO pre-trained models**
>
> We provide experiment results of fine-tuning COCO pre-trained models below. All of these models are trained on the same task samples dataset collected from RND-ALP. We found that in our settings, initializing with ImageNet supervised pre-training backbone and COCO pre-trained Mask-RCNN generally achieved similar performance.
>
> | | Train split || Test Split ||
> | :--- | :--- | :--- | :--- | :--- |
> | Method | ObjDet | InstSeg | ObjDet | InstSeg |
> | RND-ALP (ours) | 87.95 | 84.56 | 50.34 | 46.74 |
> | ImageNet Supervised | 88.75 | 85.51 | 52.78 | 46.94 |
> | COCO Supervised | 88.57 | 84.35 | 52.64 | 47.81 |
>
> Again our goal is not to outperform the performance of these models here, since their pre-training has access to a significant amount of supervisions from semantic class labels or annotations and constructing these curated datasets require huge manual efforts.

---

> ### Author Response · Authors · 2023-11-18
> **Response to Reviewer e8e8 (Part 2)**
>
> **Visual representation learning baselines**
>
> We compare to image-level visual pre-training baselines on fixed and static datasets such as ImageNet, since we propose to incorporate action signals from active embodied movements into learning visual representation in our ALP framework, which is not available when pre-training on fixed and static datasets. Given that contrastive loss is a popular self-supervised learning approach and can be integrated with a wide range of model architectures, we mainly compare to ImageNet SimCLR as the representation learning baseline model pre-trained on large-scale curated datasets. We additionally compare to CRL [1] as the previous embodied learning baseline method, and refer to ImageNet supervised pre-training in Table 3. R3M [3] utilizes language annotations of egocentric videos as another data modality that are not available in our settings, and focuses on improving success of robotics manipulation tasks, differently from our goals; thus we did not include it as a baseline in our paper draft.
>
> Importantly, we want to show that by actively exploring in embodied environments the agent can effectively learn visual representations from action information. *We are not trying to present a better pre-trained visual representation model at this point;* instead we want to deliver our key insight that leveraging learning signals from active embodied movements can provide substantial gains in learning visual perception, as also shown in Table 2 and 7.
>
> Given that more recent pre-trained visual representations are proposed, we provide reference to performance of these suggested models, including MAE [4], VC-1 [5], and R3M [3], below. We initialize the backbone encoder of Mask-RCNN models with each of pre-trained visual representations, and fine-tune on the same downstream task dataset collected from RND-ALP. We use default training configurations as ViTDet in detectron2 library [6].
>
> | | | Train split || Test Split ||
> | :--- | :--- | :--- | :--- | :--- | :--- |
> | Method | Model Architecture | ObjDet | InstSeg | ObjDet | InstSeg |
> | MAE | ViTDet w/ ViT-B | 92.09 | 88.32 | 60.26 | 53.87 |
> | VC-1 | ViTDet w/ ViT-B | 91.72 | 87.61 | 58.84 | 54.44 |
> | R3M | ResNet-50 w/ FPN | 88.66 | 84.71 | 32.27 | 26.86 |
> | RND-ALP | ResNet-50 w/ FPN | 87.95 | 84.56 | 50.34 | 46.74 |
>
> *However, our goal is not to outperform these methods here. Datasets used to train these representations, such as ImageNet and Ego4D, require large amounts of human effort for collections and annotations. This is different from our framework, where pre-training and fine-tuning datasets are autonomously collected from an active exploration agent moving with intrinsic motivations.* Specifically, R3M is trained with a video-language alignment loss on egocentric manipulation videos, which may not be especially suitable for visual semantic tasks; although MAE and VC-1 achieves much better performance, it is commonly known that ViTDet significantly outperforms on COCO benchmark [6, 7]. Again these models are trained on curated internet-scale datasets that are manually collected and annotated with huge effort. Thus, we don’t think missing comparisons to these visual representations in our paper draft would be a weakness of our framework.
>
> ---
>
> **Revisions in manuscripts**
>
> We really appreciate your detailed feedback in improving our writing. We have updated the paper draft (marked in blue), in the Introduction section to better describe the novelty and contribution of our work and in the Table 2 caption to more precisely summarize experimental results. Please let us know if this helps clarify or answer your questions!
>
> ---
>
> We would like to thank you again for your efforts and time in reviewing our manuscripts and providing substantial feedback. In the meantime please let us know whether our responses answer your questions and address your concerns and whether there are additional clarifications we could provide!
>
> ---
>
> References
>
> [1] Du, Yilun, Chuang Gan, and Phillip Isola. "Curious representation learning for embodied intelligence." Proceedings of the IEEE/CVF International Conference on Computer Vision. 2021.
>
> [2] Chaplot, Devendra Singh, et al. "Seal: Self-supervised embodied active learning using exploration and 3d consistency." Advances in neural information processing systems 34 (2021): 13086-13098.
>
> [3] Nair, Suraj, et al. "R3m: A universal visual representation for robot manipulation." arXiv preprint arXiv:2203.12601 (2022).
>
> [4] He, Kaiming, et al. "Masked autoencoders are scalable vision learners." Proceedings of the IEEE/CVF conference on computer vision and pattern recognition. 2022.
>
> [5] Majumdar, Arjun, et al. "Where are we in the search for an Artificial Visual Cortex for Embodied Intelligence?." arXiv preprint arXiv:2303.18240 (2023).
>
> [6] https://github.com/facebookresearch/detectron2/tree/main/projects/ViTDet
>
> [7] https://github.com/facebookresearch/detectron2/blob/main/MODEL_ZOO.md

---

> ### Author Response · Authors · 2023-11-21
> **A gentle reminder**
>
> Dear Reviewer e8e8,
>
> Thank you for your time and efforts in reviewing our paper draft.
>
> Based on your suggestions and feedback, we provide clarifying responses and additional experiments to answer your questions. We sincerely hope that this addresses your concerns.
>
> We would like to kindly remind that the discussion period will end soon. We wonder whether there are additional clarifications that we could further provide to address your concerns (if there are any).
>
> Thank you very much!
>
> Best, Authors

---

### Official Review · Reviewer_DHKz · 2023-11-01

**Soundness:** 4 excellent
**Presentation:** 3 good
**Contribution:** 3 good
**Rating:** 6
**Confidence:** 3

**Summary:**

THe paper proposes a two stage active learning approach to learn representations.
In the first state a agent uses intrinsic motivation as a way to explore the environment
and learn representations from the scene. That representation learned is used to learn a
downstream task on the second part.
The reward is directly based on novelty.
The paper shows improved downstream tasks results when learning while interacting in the Habitat environment.

**Strengths:**

* Interesting how they are able to use the police training gradient as the representation learning signal.
* That seems to learn a better representation which enforces better exploration at the same time.

**Weaknesses:**

1 In terms of how solid the contribution it seems in my opinion slightly marginal when compared to the related work. The inclusion of police gradient into the representation learning helps the exploration and shows a better learned representation and bigger exploration in habitat. However, I am a bit skeptical with the extent of that as a contribution. The improvement seemed not that convincing and I am honestly worried on how much that can change given variations on the environment and initialization. It is likely that different starting conditions and different scene combinations might have drastically different results.
It is expected I think for a few different polices to be trained and be able to measure the variability of the representation learned.

**Questions:**

I am curious on potential difficulties found when training with the policy gradient loss ? Was there any sort of instability ? Since it tends to different dynamics changes during the training depending on how much reward is being obtained in a given moment.

---

> ### Author Response · Authors · 2023-11-18
> **Response to Reviewer DHKz (Part 1)**
>
> We sincerely thank you for helpful feedback and insightful comments. We appreciate that our paper is recognized for several positive aspects: (1) the idea of using reinforcement learning objective for training exploration policy as part of visual representation learning approach is interesting; (2) experiments show improvements in both visual representation quality and data quality from better exploration. We address your comments and questions below:
>
> ---
>
> **Novelty**
>
> We would like to highlight that the contributions of our framework come in two perspectives.
>
> First, we train an active exploration agent from intrinsic motivation to collect diverse and informative samples for both representation learning and downstream task models. This is different from previous works, where CRL [3] only actively pre-trains visual representations or SEAL [2] only fine-tunes a detection segmentation model using actively collected labeled samples. Our experiments demonstrate that ALP benefits from both actively learned visual representations (Table 2, 3) and actively collected task samples (Table 4), showing the importance of actively collecting samples for training vision models.
>
> Second, given that the agent actively explores in visual environments, we propose a representation learning approach that *only* uses action information from embodied movements, which optimizes a reinforcement learning objective and an inverse dynamics objective. While prior works adopt similar ideas of “shared backbone” in supervised end-to-end RL tasks [4], we demonstrate that leveraging learning signals from actions can provide substantial gains in several vision tasks. This is different from popular visual representation learning methods on fixed or static datasets, such as contrastive loss [5, 6] or reconstruction error [7]. Importantly, we want to show that by actively exploring in environments the agent can effectively learn visual representations from embodied movements, as good as or better than baselines (Table 3). Our experiments demonstrate that ALP improves visual representation learning compared to self-supervised contrastive learning from the simulator (Table 2, 7) and each component contributes to better performance (Table 5).
>
> We hope this could bring attention to the importance of actively collecting data for learning visual perception, both for pre-training visual representations and for fine-tuning task-specific models. This provides a different perspective from popular large-scale curated datasets, such as ImageNet and COCO, that require large amounts of human effort for data collection and annotations; instead our exploration agent is trained actively and autonomously from intrinsic motivation. Additionally, we *only* leverage action information in embodied settings as learning signals for visual representations, which is different from popular methods trained on fixed and static datasets. Through extensive experiments and ablations on a wide range of tasks we demonstrate benefits and gains from both perspectives.
>
> ---
>
> **Results from multiple runs**
>
> We already include experimental results of multiple random seeds for RND-ALP in Table 11 in Appendix. Given the same pre-trained representation and the same task samples dataset, our 3 runs of results achieve similar performance.
>
> We additionally include downstream performance of RND-ALP over 2 runs with different runs for both pre-training and fine-tuning stages below. We generally found that performance in unseen scenarios contains slightly higher variance than that in training environments across multiple runs; overall our method could achieve consistent performance.
>
> | Train Split || Test Split ||
> | :--- | :--- | :--- | :--- |
> | ObjDet | InstSeg | ObjDet | InstSeg |
> | 87.95 | 84.56 | 50.34 | 46.74 |
> | 88.29 | 83.60 | 50.82 | 43.20 |
>
> ---
>
> **Difficulties in training instability**
>
> We mostly follow prior work [1] when running policy training experiments. We use the distributed version of policy training algorithms and parallelize more environments, in order to gather diverse samples and reduce training instabilities. We generally found that collecting more diverse rollout samples and thus using a larger batch size could help with training instability issues in our experiments. Given our available compute resources we try to parallel as many environments as possible until saturating memory constraints of machines.
>
> ---
>
> **Revisions in manuscripts**
>
> We have updated the paper draft (marked in blue), in the Introduction section to better describe the novelty and contribution of our work and in the Table 2 caption to more precisely summarize experimental results. Please let us know if this helps clarify or answer your questions!

---

> ### Author Response · Authors · 2023-11-18
> **Response to Reviewer DHKz (Part 2)**
>
> We would like to thank you again for your efforts and time in reviewing our manuscripts and providing substantial feedback. In the meantime please let us know whether our responses answer your questions and address your concerns and whether there are additional clarifications we could provide!
>
> ---
>
> References
>
> [1] Wijmans, Erik, et al. "Dd-ppo: Learning near-perfect pointgoal navigators from 2.5 billion frames." arXiv preprint arXiv:1911.00357 (2019).
>
> [2] Chaplot, Devendra Singh, et al. "Seal: Self-supervised embodied active learning using exploration and 3d consistency." Advances in neural information processing systems 34 (2021): 13086-13098.
>
> [3] Du, Yilun, Chuang Gan, and Phillip Isola. "Curious representation learning for embodied intelligence." Proceedings of the IEEE/CVF International Conference on Computer Vision. 2021.
>
> [4] Ye, Joel, et al. "Auxiliary tasks speed up learning point goal navigation." Conference on Robot Learning. PMLR, 2021.
>
> [5] Chen, Ting, et al. "A simple framework for contrastive learning of visual representations." International conference on machine learning. PMLR, 2020.
>
> [6] He, Kaiming, et al. "Momentum contrast for unsupervised visual representation learning." Proceedings of the IEEE/CVF conference on computer vision and pattern recognition. 2020.
>
> [7] He, Kaiming, et al. "Masked autoencoders are scalable vision learners." Proceedings of the IEEE/CVF conference on computer vision and pattern recognition. 2022.

---

> ### Author Response · Authors · 2023-11-21
> **A gentle reminder**
>
> Dear Reviewer DHKz,
>
> Thank you for your time and efforts in reviewing our paper draft.
>
> Based on your suggestions and feedback, we provide clarifying responses and additional experiments to answer your questions. We sincerely hope that this addresses your concerns.
>
> We would like to kindly remind that the discussion period will end soon. We wonder whether there are additional clarifications that we could further provide to address your concerns (if there are any).
>
> Thank you very much!
>
> Best, Authors

---

> > ### Comment · Reviewer_DHKz · 2023-11-22
> > **Change in evaluation**
> >
> > I do think the authors addressed my concerns and I do think the contribution is valuable.
> > Learning a representation from policy learning is a valid contribution.  My full access of novelty is limited
> > but I do think there is sufficient contribution here.
> > The concerns, also raised from other reviewers, on limited evaluation are definitely a risk
> > but I think that could be done in future work.
> >
> > Changed to minor accept.
> >
> > Regards,
> > Reviewer DHKz

---

### Official Review · Reviewer_mMb6 · 2023-11-08

**Soundness:** 4 excellent
**Presentation:** 4 excellent
**Contribution:** 3 good
**Rating:** 6
**Confidence:** 3

**Summary:**

This paper proposes a two stages framework designed to incorporate the actions of an embodied agent to learn representations from actively collected data. In the first stage, the embodied agent is actively exploring the environment driven by intrinsic motivation. The agent utilizes the action information implicitly using a shared backbone for policy and representation learning, and explicitly through inverse dynamics prediction. The second stage involves fine-tuning the learned representations for a wide range of perception downstream tasks (object detection, segmentation, and depth estimation) using random samples of the actively collected data. The authors conduct a series of experiments within simulated environments of 3D scans of real locations (the Gibson environments), demonstrating that their framework outperforms both baseline embodied learning and self-supervised learning methods, as well as pre-trained models from static datasets.

**Strengths:**

1. The paper is well-written, offering clarity and ease of understanding.
2. The framework's generality is a strong point, allowing for integration with various exploration techniques and adaptability to multiple downstream tasks.
3. Empirical evidence is compelling, especially in demonstrating the benefits of leveraging the action information through using the shared backbone (implicit) and inverse dynamics model objective (explicit).
4. Fine-tuning performance on diverse downstream perception tasks and robustness in out-of-distribution scenarios are effectively presented.
5. The exhaustive experimental comparison with other baseline embodied learning and self-supervised learning methods, alongside pre-trained models, solidifies the framework's advantages.

**Weaknesses:**

1. The scope of experiments is only limited to simulated environments of 3D scans of real locations (the Gibson environments).
2. The action space utilized in the experiments is relatively simple, including only discrete actions, no further experiments on more complex action space.
3. The novelty of this work is somewhat limited, but the empirical findings are still worthwhile.
    1. The two-stage framework is similar to the one proposed in CRL by Du et al. (2021), but using a different training objective.
    2. The proposed training objective combining the policy gradient objective and the inverse dynamics model objective is similar to Ye et al. (2021). However, Ye et al. (2021) showed only for downstream navigation tasks.

- Joel Ye, Dhruv Batra, Erik Wijmans, and Abhishek Das. Auxiliary tasks speed up learning point goal navigation. In Conference on Robot Learning, pp. 498–516. PMLR, 2021.
- Yilun Du, Chuang Gan, and Phillip Isola. Curious representation learning for embodied intelligence. In Proceedings of the IEEE/CVF International Conference on Computer Vision, pp. 10408–10417, 2021.

**Questions:**

1. Concerning the first weakness, has there been any consideration for testing with datasets from Matterport and Replica, as utilized in Chaplot et al. (2020)?
2. In relation to the second weakness, could you specify which type of embodied agent was employed in the experiments? Based on the Gibson environments, the embodied agent can be humanoid, ant, husky car, drone, minitaur, or Jackrabbot.
3. Regarding the third weakness, could you provide further clarification, if possible?
4. Regarding Table 11, which presents the variance across multiple runs, does this imply that the results were derived from a single execution of stage one, subsequently followed by various iterations of stage two? If so, could you clarify how the variance would be accounted for across multiple runs where each run includes running both the first and second stages consecutively?
5. Is there a specific rationale behind the decision to run the embodied agent for 8M frames?

- Devendra Singh Chaplot, Helen Jiang, Saurabh Gupta, and Abhinav Gupta. Semantic curiosity for active visual learning. In Computer Vision–ECCV 2020: 16th European Conference, Glasgow, UK, August 23–28, 2020, Proceedings, Part VI 16, pp. 309–326. Springer, 2020c.

---

> ### Author Response · Authors · 2023-11-18
> **Response to Reviewer mMb6 (Part 1)**
>
> We sincerely thank you for helpful feedback and insightful comments. We appreciate that our paper is recognized for several positive aspects: (1) the paper is well-written and easy to follow; (2) our framework has strong generality and can be integrated with any active exploration method; (3) we provide extensive experimental comparisons to baseline methods and empirical evidence is clear. We address your comments and questions below:
>
> ---
>
> **Limited scope of environments**
>
> While we perform all experiments in the Gibson dataset [1], different scenes in the Gibson dataset contain some extent of diversity. We follow prior work [2] in splitting training and unseen test scenes, details provided in Appendix B2. Our evaluations on unseen test scenarios could indicate some generalization capability of our framework, since these environments are not available during either pre-training or fine-tuning stages. We found it a little challenging to directly evaluate on another dataset such as Replica or Matterport3D, since these datasets capture different styles of scenes or buildings and distributions of object categories are different, which could add further complications to evaluations. As discussed in the last part, we would like to leave further extensions to other simulator datasets and/or real world environments as part of future work.
>
> ---
>
> **Simple action space**
>
> We define the action space in our experiments same as in prior works [2]. We render the Gibson dataset in the Habitat simulator [4] and train a reinforcement learning policy of a navigation agent. We tried looking up and down as possible actions similar to CRL [3], but we found that this collects weaker samples, e.g. the agent might get many observations of ceilings and floors without valid semantic annotations of objects. We thus use the same action space as SEAL [2].
> We are not aware of different choices of embodied agents as you described under our setup; could you further clarify this part? Thanks! Although we currently utilize a simple action space in our experiments, we already observe performance gains, indicating effectiveness of leveraging action information for visual representation learning. As discussed in the last part, we would like to leave further investigations on more complex action space as part of future work.
>
> ---
>
> **Novelty**
>
> We would like to highlight that the contributions of our framework come in two perspectives.
>
> First, we train an active exploration agent from intrinsic motivation to collect diverse and informative samples for both representation learning and downstream task models. This is different from previous works, where CRL [3] only actively pre-trains visual representations or SEAL [2] only fine-tunes a detection segmentation model using actively collected labeled samples. Our experiments demonstrate that ALP benefits from both actively learned visual representations (Table 2, 3) and actively collected task samples (Table 4), showing the importance of actively collecting samples for training vision models.
>
> Second, given that the agent actively explores in visual environments, we propose a representation learning approach that *only* uses action information from embodied movements, which optimizes a reinforcement learning objective and an inverse dynamics objective. While prior works adopt similar ideas of “shared backbone” in supervised end-to-end RL tasks [5], we demonstrate that leveraging learning signals from actions can provide substantial gains in several vision tasks. This is different from popular visual representation learning methods on fixed or static datasets, such as contrastive loss [6, 7] or reconstruction error [8]. Importantly, we want to show that by actively exploring in environments the agent can effectively learn visual representations from embodied movements, as good as or better than baselines (Table 3). Our experiments demonstrate that ALP improves visual representation learning compared to self-supervised contrastive learning from the simulator (Table 2, 7) and each component contributes to better performance (Table 5).
>
> We hope this could bring attention to the importance of actively collecting data for learning visual perception, both for pre-training visual representations and for fine-tuning task-specific models. This provides a different perspective from popular large-scale curated datasets, such as ImageNet and COCO, that require large amounts of human effort for data collection and annotations; instead our exploration agent is trained actively and autonomously from intrinsic motivation. Additionally, we *only* leverage action information in embodied settings as learning signals for visual representations, which is different from popular methods trained on fixed and static datasets. Through extensive experiments and ablations on a wide range of tasks we demonstrate benefits and gains from both perspectives.

---

> ### Author Response · Authors · 2023-11-18
> **Response to Reviewer mMb6 (Part 2)**
>
> **Results from multiple runs**
>
> In addition to Table 11, we include downstream performance of RND-ALP over 2 runs with different runs for both pre-training and fine-tuning stages below. We generally found that performance in unseen scenarios contains slightly higher variance than that in training environments across multiple runs; overall our method could achieve consistent performance.
>
> | Train Split || Test Split ||
> | :--- | :--- | :--- | :--- |
> | ObjDet | InstSeg | ObjDet | InstSeg |
> | 87.95 | 84.56 | 50.34 | 46.74 |
> | 88.29 | 83.60 | 50.82 | 43.20 |
>
> ---
>
> **Number of pre-training steps**
>
> We try to follow prior work [3] to pretrain for 10M steps, but given available compute resources we reduce the number of parallel environments than in their implementation and our exploration policy is trained for around 8M steps. We fix this parameter value across experiments to compare effects of other components of our framework on final performance.
>
> ---
>
> **Revisions in manuscripts**
>
> We have updated the paper draft (marked in blue), in the Introduction section to better describe the novelty and contribution of our work and in the Table 2 caption to more precisely summarize experimental results. Please let us know if this helps clarify or answer your questions!
>
> ---
>
> We would like to thank you again for your efforts and time in reviewing our manuscripts and providing substantial feedback. In the meantime please let us know whether our responses answer your questions and address your concerns and whether there are additional clarifications we could provide!
>
> ---
>
> References
>
> [1] Xia, Fei, et al. "Gibson env: Real-world perception for embodied agents." Proceedings of the IEEE conference on computer vision and pattern recognition. 2018.
>
> [2] Chaplot, Devendra Singh, et al. "Seal: Self-supervised embodied active learning using exploration and 3d consistency." Advances in neural information processing systems 34 (2021): 13086-13098.
>
> [3] Du, Yilun, Chuang Gan, and Phillip Isola. "Curious representation learning for embodied intelligence." Proceedings of the IEEE/CVF International Conference on Computer Vision. 2021.
>
> [4] Savva, Manolis, et al. "Habitat: A platform for embodied ai research." Proceedings of the IEEE/CVF international conference on computer vision. 2019.
>
> [5] Ye, Joel, et al. "Auxiliary tasks speed up learning point goal navigation." Conference on Robot Learning. PMLR, 2021.
>
> [6] Chen, Ting, et al. "A simple framework for contrastive learning of visual representations." International conference on machine learning. PMLR, 2020.
>
> [7] He, Kaiming, et al. "Momentum contrast for unsupervised visual representation learning." Proceedings of the IEEE/CVF conference on computer vision and pattern recognition. 2020.
>
> [8] He, Kaiming, et al. "Masked autoencoders are scalable vision learners." Proceedings of the IEEE/CVF conference on computer vision and pattern recognition. 2022.

---

> ### Author Response · Authors · 2023-11-21
> **A gentle reminder**
>
> Dear Reviewer mMb6,
>
> Thank you for your time and efforts in reviewing our paper draft.
>
> Based on your suggestions and feedback, we provide clarifying responses and additional experiments to answer your questions. We sincerely hope that this addresses your concerns.
>
> We would like to kindly remind that the discussion period will end soon. We wonder whether there are additional clarifications that we could further provide to address your concerns (if there are any).
>
> Thank you very much!
>
> Best, Authors

---

> > ### Comment · Reviewer_mMb6 · 2023-11-22
> > **No further questions**
> >
> > Dear Authors,
> >
> > Thank you for your comprehensive and detailed reply.
> >
> > I have thoroughly read your response and find that all of my questions have been addressed.
> >
> > Best, Reviewer mMb6

---

### Official Review · Reviewer_h74j · 2023-11-09

**Soundness:** 3 good
**Presentation:** 4 excellent
**Contribution:** 2 fair
**Rating:** 3
**Confidence:** 4

**Summary:**

The authors demonstrate a system that actively learns visual representation and collects data to learn from. Learning signals are RL reward loss and an additional loss from Inverse Dynamics Prediction - which comes from predicting actions given a sequence of frames. It is shown that the system outperforms baseline methods.

**Strengths:**

* The system convincingly outperforms the baseline methods.
* The ablation experiment adds to the strength of the method

**Weaknesses:**

* While the results seems strong, the work severely lacks novelty. The two points of novelty as reported are - i. Combined stages for collecting frames and learning from it. and ii. Sharing the backbone for policy and representation learning. In my opinion, these are not significant changes.
* All the experiments are done on a single environment (Gibson)
* Not clear why Inverse Dynamics Prediction is a good learning signal.

**Questions:**

1. Have you tried some more recent methods for self-sup training like MAE or VideoMAE?
2. What is the rationale behind IDM? It seems without reason to me, since the RL reward should already be a learning signal to associate visual frames to action.

---

> ### Author Response · Authors · 2023-11-18
> **Response to Reviewer h74j (Part 1)**
>
> We sincerely thank you for helpful feedback and insightful comments. We appreciate that our paper is recognized for several positive aspects: (1) our experimental results outperform baselines in multiple tasks; (2) our ablation experiments effectively strengthen our framework. We address your comments and questions below:
>
> ---
>
> **Novelty**
>
> We would like to highlight that the contributions of our framework come in two perspectives.
>
> First, we train an active exploration agent from intrinsic motivation to collect diverse and informative samples for both representation learning and downstream task models. This is different from previous works, where CRL [2] only actively pre-trains visual representations or SEAL [1] only fine-tunes a detection segmentation model using actively collected labeled samples. Our experiments demonstrate that ALP benefits from both actively learned visual representations (Table 2, 3) and actively collected task samples (Table 4), showing the importance of actively collecting samples for training vision models.
>
> Second, given that the agent actively explores in visual environments, we propose a representation learning approach that *only* uses action information from embodied movements, which optimizes a reinforcement learning objective and an inverse dynamics objective. While prior works adopt similar ideas of “shared backbone” in supervised end-to-end RL tasks [3], we demonstrate that leveraging learning signals from actions can provide substantial gains in several vision tasks. This is different from popular visual representation learning methods on fixed or static datasets, such as contrastive loss [4, 5] or reconstruction error [6]. Importantly, we want to show that by actively exploring in environments the agent can effectively learn visual representations from embodied movements, as good as or better than baselines (Table 3). Our experiments demonstrate that ALP improves visual representation learning compared to self-supervised contrastive learning from the simulator (Table 2, 7) and each component contributes to better performance (Table 5).
>
> We hope this could bring attention to the importance of actively collecting data for learning visual perception, both for pre-training visual representations and for fine-tuning task-specific models. This provides a different perspective from popular large-scale curated datasets, such as ImageNet and COCO, that require large amounts of human effort for data collection and annotations; instead our exploration agent is trained actively and autonomously from intrinsic motivation. Additionally, we *only* leverage action information in embodied settings as learning signals for visual representations, which is different from popular methods trained on fixed and static datasets. Through extensive experiments and ablations on a wide range of tasks we demonstrate benefits and gains from both perspectives.
>
> ---
>
> **Inverse dynamics prediction as training objective**
>
> While reinforcement learning objective provides a learning signal to associate visual frames to action, we additionally make an observation that consecutive frames encode agent’s movements in interactive settings. We design our inverse dynamics model to predict sequences of actions given observations, as shown in Figure 1. In this case, we directly infuse action information into visual representation learning, in addition to policy and reward information from reinforcement learning objectives. Our inverse dynamics model also provides temporal learning signals since it is trained on sequence of trajectories over multiple steps, different from RL policy that associate observation and action at every single step. Our ablation experiments in Table 5 clearly show that both of these components contribute to improved performance.

---

> ### Author Response · Authors · 2023-11-18
> **Response to Reviewer h74j (Part 2)**
>
> **Visual representation learning baselines**
>
> We compare to image-level visual pre-training baselines on fixed and static datasets such as ImageNet, since we propose to incorporate action signals from active embodied movements into learning visual representation in our ALP framework, which is not available when pre-training on fixed and static datasets. Given that contrastive loss is a popular self-supervised learning approach and can be integrated with a wide range of model architectures, we mainly compare to ImageNet SimCLR as the representation learning baseline model pre-trained on large-scale curated datasets. We additionally compare to CRL [2] as the previous embodied learning baseline method, and refer to ImageNet supervised pre-training in Table 3.
>
> Importantly, we want to show that by actively exploring in embodied environments the agent can effectively learn visual representations from action information. *We are not trying to present a better pre-trained visual representation model at this point;* instead we want to deliver our key insight that leveraging learning signals from active embodied movements can provide substantial gains in learning visual perception, as also shown in Table 2 and 7.
>
> Given that more recent pre-trained visual representations are proposed, we provide reference to performance of these suggested models, including MAE [6], below. We initialize the backbone encoder of Mask-RCNN models with each of pre-trained visual representations, and fine-tune on the same downstream task dataset collected from RND-ALP. We use default training configurations as ViTDet in detectron2 library [7].
>
> | | | Train split || Test Split ||
> | :--- | :--- | :--- | :--- | :--- | :--- |
> | Method | Model Architecture | ObjDet | InstSeg | ObjDet | InstSeg |
> | MAE | ViTDet w/ ViT-B | 92.09 | 88.32 | 60.26 | 53.87 |
> | RND-ALP | ResNet-50 w/ FPN | 87.95 | 84.56 | 50.34 | 46.74 |
>
> *However, our goal is not to outperform these methods here. Datasets used to train these representations, such as ImageNet, require large amounts of human effort for collections and annotations.* This is different from our framework, where pre-training and fine-tuning datasets are autonomously collected from an active exploration agent moving with intrinsic motivations. Specifically, although MAE achieves much better performance, it is commonly known that ViTDet significantly outperforms on COCO benchmark [7, 8]. Again these models are trained on curated internet-scale datasets that are manually collected and annotated with huge effort. Thus, we don’t think missing comparisons to these visual representations in our paper draft would be a weakness of our framework.
>
> ---
>
> **Revisions in manuscripts**
>
> We have updated the paper draft (marked in blue), in the Introduction section to better describe the novelty and contribution of our work and in the Table 2 caption to more precisely summarize experimental results. Please let us know if this helps clarify or answer your questions!
>
> ---
>
> We would like to thank you again for your efforts and time in reviewing our manuscripts and providing substantial feedback. In the meantime please let us know whether our responses answer your questions and address your concerns and whether there are additional clarifications we could provide!
>
> ---
>
> References
>
> [1] Chaplot, Devendra Singh, et al. "Seal: Self-supervised embodied active learning using exploration and 3d consistency." Advances in neural information processing systems 34 (2021): 13086-13098.
>
> [2] Du, Yilun, Chuang Gan, and Phillip Isola. "Curious representation learning for embodied intelligence." Proceedings of the IEEE/CVF International Conference on Computer Vision. 2021.
>
> [3] Ye, Joel, et al. "Auxiliary tasks speed up learning point goal navigation." Conference on Robot Learning. PMLR, 2021.
>
> [4] Chen, Ting, et al. "A simple framework for contrastive learning of visual representations." International conference on machine learning. PMLR, 2020.
>
> [5] He, Kaiming, et al. "Momentum contrast for unsupervised visual representation learning." Proceedings of the IEEE/CVF conference on computer vision and pattern recognition. 2020.
>
> [6] He, Kaiming, et al. "Masked autoencoders are scalable vision learners." Proceedings of the IEEE/CVF conference on computer vision and pattern recognition. 2022.
>
> [7] https://github.com/facebookresearch/detectron2/tree/main/projects/ViTDet
>
> [8] https://github.com/facebookresearch/detectron2/blob/main/MODEL_ZOO.md

---

> ### Author Response · Authors · 2023-11-21
> **A gentle reminder**
>
> Dear Reviewer h74j,
>
> Thank you for your time and efforts in reviewing our paper draft.
>
> Based on your suggestions and feedback, we provide clarifying responses and additional experiments to answer your questions. We sincerely hope that this addresses your concerns.
>
> We would like to kindly remind that the discussion period will end soon. We wonder whether there are additional clarifications that we could further provide to address your concerns (if there are any).
>
> Thank you very much!
>
> Best, Authors

---

### Meta-Review · Area_Chair_1Nri · 2023-12-09

**Metareview:**

**Summary**

The paper introduces an embodied learning framework designed to enhance visual perception learning. The method involves actively exploring complex 3D environments and incorporates action information into representation learning. This is achieved through optimizing a reinforcement learning policy and an inverse dynamics prediction objective. The framework, ALP, aims to overcome the limitations of training on fixed datasets by adapting to evolving environments and data. It is shown to outperform existing baselines in downstream perception tasks, demonstrating greater robustness and adaptability in various scenarios.

**Strength**

The paper's strengths lie in its approach to learning visual representations through active exploration and action-aware training objectives. It effectively integrates different exploration methods and adapts to multiple perception tasks. Empirical evidence supports the framework's effectiveness, particularly in out-of-distribution scenarios. The experimental comparisons provide a foundation for understanding the framework's benefits over baseline methods. The paper is clearly written and well-organized.

**Weakness**

The paper's scope is limited to simulated environments, specifically the Gibson environments. The action space used in the experiments is relatively simple and lacks complexity, which might limit the diversity of the dataset used for training. While the paper provides novel insights, its contributions are somewhat incremental and resemble existing methods like CRL. Also, the paper underperforms compared to some baselines in certain tasks, raising questions about its scalability and generalizability. Further comparisons with recent representation learning methods and different data collection strategies are needed for a more comprehensive evaluation.

**Justification For Why Not Higher Score:**

The reviewers acknowledged the authors' responses, with some updating their evaluation to a more positive stance, appreciating the incorporation of policy gradient into representation learning. However, some reviewers remained unconvinced about the scalability and generalizability of the method. They pointed out that while the paper provided interesting insights, the results, especially in comparison to offline pre-trained models, didn't convincingly demonstrate the superiority of actively learned representations. These concerns indicated a need for broader and more varied evaluations to fully establish the framework's efficacy.

**Justification For Why Not Lower Score:**

N/A

---

### Decision · Program_Chairs · 2024-01-16

Reject